# Phenotypic Screening and Marker-Assisted Validation of Sources of *Aphis craccivora* Koch Resistance in Cowpea (*Vigna unguiculata* L.)

**DOI:** 10.3390/ijms26094406

**Published:** 2025-05-06

**Authors:** Galalea Gillian Gaonosi, Lekgari Lekgari, Marang Mosupiemang, Metseyabeng Nametso Sehularo, Aobakwe Boisy Tshegofatso, Bamphithi Tiroesele, Tiny Motlhaodi, Samodimo Ngwako, Motlalepula Pholo-Tait

**Affiliations:** 1Department of Crop and Soil Sciences, Botswana University of Agriculture and Natural Resources, Gaborone 00267, Botswana; gmmolutsi@buan.ac.bw (G.G.G.); marangmosupiemang@yahoo.com (M.M.); 201200327@buan.ac.bw (M.N.S.); atshegof@buan.ac.bw (A.B.T.); btiroese@buan.ac.bw (B.T.); sngwako@buan.ac.bw (S.N.); 2Department of Field Crops and Horticulture, National Agricultural Research Development Institute, Gaborone 00267, Botswana; lekgari@nardi.org.bw (L.L.); tiny@nardi.org.bw (T.M.); 3Centre for Bioeconomy, Faculty of Research and Graduate Studies, Botswana University of Agriculture and Natural Resources, Gaborone 00267, Botswana

**Keywords:** *Aphis craccivora*, *Vigna unguiculata*, SNP1_0912, CP 171F/172R, quantitative trait loci, screening, resistance, susceptible, tolerance

## Abstract

*Aphis craccivora* significantly affects cowpea (*Vigna unguiculata* L.) production, leading to yield reductions. Management strategies encompass physical barriers and biological and chemical methods, which can be costly and detrimental to the environment. Host-plant resistance offers a more sustainable alternative. This study evaluated cowpea genotypes in a screenhouse experiment. Tswana and B261-B were resistant, while B301, B259, and ER7 showed a tolerance phenotype. Tswana exhibited a low aphid population and minimal plant damage, probably due to suppression of reproduction and fecundity. Conversely, IT97K-556-6, SARI-21KTA-6, SARC 1-57-2, B013-F, B339, and Blackeye were susceptible to aphids, as shown by high aphid populations and dense sooty molds. Severe damage to plant vigor may be linked to direct aphid feeding and reduced photosynthesis efficiency. SNP1_0912 and CP 171F/172R markers confirmed aphid resistance in Tswana and ER7 as well as in the IT97k-556-6 and SARI-21KTA-6 controls. The inverse susceptible phenotype in the control group suggests that the markers may not function properly due to negative interactions between quantitative trait loci (QTL) and environmental factors. This could also indicate the presence of different aphid biotypes that severely damage Western African breeding lines. This study offers essential insights for breeding aphid-resistant cowpea varieties. Future efforts will involve sequencing Tswana to identify more resistance sources and create novel markers.

## 1. Introduction

Cowpea (*Vigna unguiculata* L. Walp.) (2n = 2x = 22) is a member of the *Phaseoleae* tribe of the Leguminosae family. It belongs to the Fabaceae family and Faboideae sub-family [1]. It is a source of protein, minerals, and vitamins in thousands of low-income households’ diets [1]. Statistics Botswana indicated that cowpea is the third most important crop after maize (*Zea mays* L.) and grain sorghum (*Sorghum bicolor* (L.) Moench) in Botswana [2]. It is a major source of dietary protein that nutritionally complements staple low-protein cereal [3]. The cowpea grain is a low-cost alternative to more expensive fish and cattle products on the African market [4]. The nutrients from cowpea are obtained from various parts such as dry seeds, green pods, green seeds, and tender green leaves. It is a vital cash crop that contributes to farmers’ food security and income generation [4]. In addition to its nutritional value, cowpea is an essential component of the cropping systems in semi-arid regions of Sub-Saharan Africa [5,6]. This has been attributed to its ability to fix atmospheric nitrogen, which improves soil fertility for the success of cereal crops grown in rotation, particularly in smallholder farming systems. In this respect, cowpea is a high-potential strategic crop for addressing the complex challenges of hunger, malnutrition, environmental sustainability, climate change, and rising food prices, which the global community will confront in the coming decades [7].

The Southern African region, including Botswana, Namibia, Zambia, Zimbabwe, Mozambique, and the Republic of South Africa, is considered the centre of diversity of *V. unguiculata* [8]. In Botswana, diverse cowpea germplasm collections are conserved in the National Plant Genetic Resource Centre (NPGRC)-Ministry of Agriculture, and their duplicates are conserved in the Southern African Development Community (SADC) gene bank in Lusaka/Zambia [9,10]. The Botswana National Plant Genetic Resource Centre currently has a collection of over 1487 accessions of Vigna unguiculata [9]. Despite the large diversity of conserved genotypes, the genetic diversity within cowpeas remains relatively narrow, which further contributes to the large prevalence of low-yielding genotypes [11].

In addition, the yield potential and the stability of the grain quality of cowpeas are further restricted by the prevalence of abiotic and biotic stresses. Amongst biotic stresses, insects are considered primarily responsible for 90–100% of the yield reduction [12]. *Aphis craccivora* Koch (Homoptera: Aphididae), commonly known as cowpea aphid, is a cowpea pest of economic importance, causing significant 50–60% yield losses in Botswana [13]. Global reports indicated yield losses of 35% in Africa, and up to 20% to 40% yield losses were reported in Asia [14]. Cowpea aphids primarily infest the succulent parts of the plant, such as young shoots, leaves, and pods [15]. The aphid feeding damage includes sucking and removing plant sap, which reduces the nutrients and water available to the crop [16]. Aphid feeding eventually leads to chlorosis and stunting, which delays the commencement of flowering, and this can cause plant mortality when infestations are substantial, especially during the seedling stage [17]. Aphids excrete a sugary plant substance known as honeydew, and black sooty-like fungus (sooty mold) can develop on honeydew, reducing photosynthesis. This occurs by physically blocking sunlight from reaching the leaf surface, thereby hindering the plant’s ability to produce food through the photosynthetic process [18]. Besides direct plant damage, aphids may transmit viruses such as Bean common mosaic virus (BcMV) and Pea seed-borne mosaic virus (PsbMV) [19]. The aphid’s ability to reproduce rapidly and its potential to develop resistance to chemical pesticides makes it difficult to control [20]. When there is less food and the environment becomes less favourable due to overcrowding, winged forms occur, further expanding their territory and causing more damage [21].

Several management approaches, including physical barriers and biological or chemical methods, are employed to control aphids in cowpeas [22]. It was previously reported [16] that these approaches are costly, labour-intensive, and may result in environmental biases. Host-plant resistance (HPR), which describes the plant’s innate resistance usually controlled by one or more genes, offers a sustainable and environmentally friendly approach to aphid management [23]. This pest management approach enables plants to withstand and survive significant pest damage due to their genetically inherited traits [24]. Developing host-plant pest-resistance varieties has become a crucial component in an integrated crop management system [25]. The concomitant aphid infestation screening and marker-assisted selection (MAS) have become the most promising tools for breeding, using molecular markers to identify and select genes or genomic regions of interest without environmental bias [26,27]. Meanwhile, phenotypic aphid infestation culture screening studies reported that the Blackeye commercial variety was highly susceptible to aphids due to lower fecundity compared to the Tswana cowpea variety [28]. In another study, landraces B383, B261-B, and Tswana exhibited the longest pre-reproductive periods, reduced aphid population growth, and reproductive success, suggesting an aphid resistance phenotype [29]. This was further confirmed in another study that demonstrated reduced aphid reproductive growth rates in B383 and B261-B landraces, while the inverse was observed for landraces B013-F, B339, and the Blackeye variety [30]. The findings were only based on the extended pre-reproductive reduction in population growth rate, survival, and fecundity [30]. However, the genetic mode of inheritance linked to aphid resistance, whether through simple inherited trait loci (SITLs) or quantitative trait loci (QTLs), remains unclear.

Considerable progress in West Africa has been made in the past years in developing cowpea varieties resistant to aphids using a marker-assisted selection strategy [31], confirming resistance to cowpea aphids in SARC 1-57-2 accession in Ghana. Resistance to aphids happens through antibiosis, which is controlled by a single dominant gene [32,33]. The aphid genes from SARC 1-57-2 were confirmed using a codominant SSR marker, CP 171F/172R, which is several centiMorgans in genetic distance from the resistance gene locus [34]. Genetic mapping of the aphid resistance in IT97K-556-6, an accession from Ghana, revealed two QTLs on cowpea linkage groups LG1 (minor) and LG7 (major), both with favourable alleles contributed from IT97K-556-6. The SNP1_0912 marker flanking the major aphid resistance QTL (QAc-vu7.1) was used to confirm a QTL conferring resistance to cowpea aphid in IT97K-556-6. This marker is located at 21.7 cM, and the QTL is at 22 cM on chromosome 2 of the cowpea genome [33]. In KVX295-2-124-99, two duplicate genes located on chromosomes 3 and 7 were reported to be involved in the aphid resistance ability, as demonstrated by segregating populations using marker MA61, which is located on chromosome 3, and marker MA70, on chromosome 7 [35]. Given the molecular variations in cowpea biotypes across different geographical locations in West Africa, this study aimed to evaluate cowpea’s response to aphid infestation and validate sources of resistance to aphids that could serve as parental lines in a breeding program.

## 2. Results

### 2.1. Cowpea Adult Aphid Population Density Growth

The number of adult aphids on various genotypes was insignificant for all days except for days 11, 13, and 15 (Table 1). Landrace B261-B showed the significantly highest number of adult aphids (70.2) on day 11 after infestation. It was followed by B301 and IT97K-556-6, with an average number of adult aphids of 64.8 and 49.3, respectively. IT97K-556-6 demonstrated the significantly highest number of adult aphids (192.2) compared to other genotypes on day 13 after infestation. In addition to IT97K-556-6, a significantly higher number of adult aphids was observed in SARC 1-57-2 (189), B261-B (147.2), B339 (121.8), and Blackeye (115.7). The results demonstrated that Tswana had the lowest number of aphids (57.7) on day 15, which was significantly lower than those in B261-B (251.7), SARI-2KTA-6 (244.0), IT97K-556-6 (255.3), and KVX295-2-124-99 (377.8). The latter showed the highest number of adult aphids compared to other genotypes. Generally, most genotypes, including Tswana, ER7, B013-F, B261-B, B301, B359, SARI-2KTA-6, and IT97K-556-6, reached their peak number of adult aphids at day 17. However, SARC 1-57-2, KVX295-2-124-99, and B339 reached peak adult aphid counts on day 15 after infestation. Only the Blackeye variety attained the peak number of adult aphids 19 days after infestation.

### 2.2. Number of Nymphs Produced

The reproduction rate of the number of nymphs significantly differed among genotypes on days 7, 9, 17, and 23 (Table 2). B301 showed the significantly highest number of nymphs, which was 64.2 and 149.3 on days 7 and 9, respectively. On day 17, Tswana had the lowest number of nymphs (248.5), and this was significantly lower than those observed for B261-B (1672.3), B359 (1672.3), SARI-2KTA-6 (1184.3), and IT97K-556-6 (1636.3). However, the significantly highest number of nymphs (1672.3) was shown on both B261-B and B359 landraces. Intriguingly, genotypes B013-F, B261-B, B301, B359, SARI-2KTA-6, SARC 1-57-2, and IT97K-556-6 reached the peak number of nymphs on day 17, which ranged between 981.5 and 1672.3. Genotype KVX295-2-124-99 took longer (23 days) to reach its peak number of nymphs compared to other genotypes. On day 23, IT97K-556-6 had a significantly higher number of nymphs (1573.5) than all other genotypes except for Blackeye and KVX295-2-124-99. Conversely, the significantly lowest number of nymphs was observed on day 23 in Tswana (267.7), which was significantly lower than in the Blackeye, IT97K-556-6, and KVX295-2-124-99 genotypes. There was a general decline in the number of nymphs produced from day 23 in all genotypes, although the decline was insignificant amongst genotypes on day 23 and day 27.

### 2.3. The Number of Alates (Winged Aphids)

Generally, the peak number of alates for most genotypes was reached on day 23 after infestation. Despite a non-significant variation in the number of aphids with wings on all other days, a significant variation was observed on day 23 (Table 3). On this day, Blackeye had the highest number of alates (219), which was significantly higher than those observed in Tswana (48.7), B261-B (53.5), and B359 (83.7). Despite a decreased trend in the number of alates after day 23, SARI-2KTA-6 and KVX295-2-124-99 showed an increased number of alates on day 27. The number of alates in those genotypes was significantly higher than in B013-F, B261-B, and IT97K-556-6.

### 2.4. Aphid Damage and Plant Vigour Score

Figure 1 shows the plant damage and vigour following aphid infestation. Scoring was performed according to the method described in previous studies [14,36]. The scoring was based on a scale from 1 to 5, where ratings of 1 to 2 indicate some resistance, a rating of 3 indicates tolerance, and ratings of 4 to 5 represent susceptible cultivars. The local commercial varieties Tswana and landrace B261-B demonstrated resistant phenotypes (Figure 2A,B), with score ratings of 2.0 and 2.67, respectively (Figure 1). Furthermore, tolerant phenotypes were observed in the B301 and B359 landraces (Figure 2C,D), both of which scored 3 (Figure 1), as well as in the variety ER7, which scored 3.2 (Figure 1 and Figure 2E). The breeding line KVX295-2-124-99 received a plant vigour score of 3.5, indicating an aphid-tolerant phenotype (Figure 1 and Figure 2F).

Conversely, Blackeye and SARC 1-57-2 received the highest damage score ratings, 4.73 and 4.67, respectively (Figure 1), indicating that they are susceptible phenotypes (Figure 3A). Additionally, the susceptible genotypes included both B013-F and B339 landraces, which received a rating of 4.17. The breeding lines IT97K-556-6 and SARI-2KTA-6 also exhibited susceptible phenotypes, with scores of 4.0 and 4.4, respectively. 

### 2.5. Genotypic Marker-Assisted Screening for Aphid Resistance

SNP1_0912 marker flanking resistance to aphid locus, with an expected aphid resistance band of ~165 bp and a susceptible band of ~150 bp, was used to segregate resistant and susceptible genotypes. The homozygous ~165 bp resistant band of interest was amplified in Tswana (Figure 4A) and ER7 (Figure 4C). However, the heterozygous ~165 bp resistant band and the ~150 bp susceptible band were amplified in Blackeye (Figure 4B). The SNP1_0912 marker also amplified a ~165 bp resistant band in ER7 (Figure 4C). Furthermore, the CP 171f/172r marker, which confers aphid resistance in sari-21kta-6, further confirmed homozygous aphid resistance in Tswana (Figure 4D) and ER7 (Figure 4E).

## 3. Discussion

Reproduction in *Aphis craccivora* happens through parthenogenesis, allowing populations to increase rapidly and making infestations difficult to control. High populations result from the aphids’ prolific reproduction within a short period, and nymphs mature into reproductive adults, leading to a rapid population growth of aphids [37]. This was evident in this study, as the population of adults and nymphs rapidly increased 11 days after infestation, peaking between days 17 and 19. The general rapid increase in the number of nymphs and adult aphids could be attributed to the rise in temperature during the experimental period. High-temperature conditions have been reported to be associated with a rapid increase in colony expansion, especially during their larval stages and the time to reproductive maturity [30]. Meanwhile, previous studies reported a rapid increase in aphid populations between 7 and 10 days after infestation [38], while a high density of aphids was observed 14 days after infestation across most accessions [14]. In the current study, the earlier observed increase in aphid population may have resulted from high temperatures during the growth period and the use of different cowpea accessions, which could have had varying genetic makeup. Due to the high aphid population and probably food shortages, winged aphid offspring increased after day 17, suggesting the migration of aphids to better food sources [37]. While increased aphid density promoted wing formation, the diminishing food sources could have contributed to environmentally induced dimorphism (polyphenism) among parthenogenetic females [39].

The phenotypic screening for aphid resistance showed that Tswana had fewer adult aphid and nymph infestations than all genotypes. The plants exhibited slight signs of damage, as indicated by a slight yellowing of the lower leaves without any capping. Similar results were observed in a previous study when Tswana was compared to Blackeye [29]. According to the aphid resistance scores, the less damaged plant vigour and the low aphid infestation in Tswana demonstrated a moderately resistant phenotype [40]. Resistance to aphids in Tswana could be attributed to the antibiosis mechanism, which involves the production of poisonous metabolites, possibly through suppressed reproductive rate and fecundity [41]. In addition to Tswana, landrace B261-B exhibited a resistant phenotype despite the high aphid density. This was in agreement with previous studies, which reported that seedlings from cowpea-resistant varieties grew vigorously and survived despite aphid infestation [42]. Therefore, resistance to a high aphid population might be attributed to antixenosis, possibly through the suppression of aphid feeding intensity [40] and the probable redistribution of nutrients from damaged tissues, thereby reducing the impact of aphid feeding on overall plant growth [43].

Consistent with the previous study [30], landraces B301 and B359 were tolerant to aphid damage, exhibiting yellowing of the lower leaves and slight capping. Tswana and B359 showed comparable nymphs and adult aphids from 23 to 27 days after aphid infestation, as previously reported [29]. However, despite a comparable aphid population, the severity of plant damage resulted in Tswana being moderately resistant to aphid plant damage, while B359 showed tolerance. The difference could be attributed to damage caused by an increased population of nymphs on day 17 after aphid infestation in B359, as compared to a consistent resistance level achieved through suppression of the extended pre-reproductive period and a reduction in population growth rate in Tswana [30].

Consistent with a previous study [30], B301 and B359 were tolerant to aphid damage, exhibiting yellowing of the lower leaves and slight capping. Additionally, ER7 and KVX295-2-124-99 were also resistant to aphid damage. Tswana and B359 showed comparable nymphs and adult aphids from 23 to 27 days after aphid infestation, as previously reported [29]. However, despite a comparable aphid population, the severity of plant damage resulted in Tswana being moderately resistant to aphid plant damage, while B359 showed tolerance. The difference could have been attributed to damage caused by an increased population of nymphs on day 17 after aphid infestation in B359, as compared to a consistent resistance level achieved through suppression of the extended pre-reproductive period and a reduction in population growth rate in Tswana [30]. Considering their differences in the level of resistance against aphids, these genotypes could be less favourable/susceptible to aphid infestation, probably through antibiosis and antixenosis compounds that can disturb the functioning, growth, and development of aphids [44,45]. The possible antibiosis-related response to aphid damage could be attributed to the suppression of reproduction, as reflected by the lower number of adults, which subsequently showed a low reproduction rate, resulting in low fecundity (fewer nymphs). On the other hand, the lower damage to plant vigour and low plant mortality rate might have been due to antixenosis, probably through prevention or the reduction of the duration of aphid’s stylet penetration beyond the epidermis and mesophyll [46,47]. This could have contributed to the decrease in aphid-salivary-related virus transmission. Beyond the aphid feeding effect, the decreased reproductive rate and fecundity might have been attributed to induced intricate aphid–host interaction mechanisms, including aphid feeding-induced phytohormones-mediated plant defence against aphids, as reported in various species-specific aphids [48,49,50]. Another possible mechanism could involve the detection of aphid salivary effectors by R proteins, initiating a stronger effector-triggered immunity through the induction of transcriptional factors. The upregulation of transcriptional factors would have subsequently induced the phytohormones and secondary metabolites [51].

On the contrary, Blackeye, B013-F, B339, IT97K-556-6, SARC1-57-2, and SARI-2KTA-6 were susceptible to aphids in the current study. The aphid susceptibility of Blackeye was positively associated with high aphid reproduction and high fecundity, which depleted food sources as reflected by an increased number of alates [21]. In IT97K-556-6, the high population of aphids resulted in damage to the plants, contradicting previously reported tolerance to aphids, which had been linked to a 40% survival rate of seedlings up to 21 days after infestation, even in the presence of a large number of aphids [38]. This was inconsistent with the reported highest level of antibiosis, which was attributed to the slowing of aphid development and multiplication [52,53,54]. The prevalence of the sooty mold in B013-F, B339, IT97K-556-6, SARC1-57-2, and SARI-2KTA-6 might have substantially damaged the plant’s vigour beyond aphid damage. The sooty mold might have interfered with photosynthesis, thereby suppressing the plant’s ability to effectively regulate light reactions in chloroplasts and disrupting the retrograde signalling necessary for the conversion of proplastids to chloroplasts. As a result, this might have led to inadequate energy supply for downstream metabolism and overall plant growth [55]. Furthermore, the severe wilting of the plants suggested the transmission and rapid spread of the plant virus within the plant beyond the direct feeding effect. This could have contributed to symptoms such as yellowing, wilting, and stunted growth, which ultimately caused more harm to the plants than the direct feeding of the aphids alone [56]. Meanwhile, variations in resistance levels to aphids have been previously reported among different lines in various regions of West Africa [54]. For instance, the stability of aphid resistance in IT97K-499-35 was reported in Nigeria and Burkina Faso [54]; however, it was susceptible to aphids in Ghana [12]. The high aphid infestation and sooty mold may be linked to a more virulent biotype of aphids in Botswana. This biotype could be damaging crops in West African varieties that contain host-plant resistance genes.

After aphid screening, DNA markers associated with genes that confer resistance in IT97K-556-6 and SARI-2KTA-6 were used to validate the sources of resistance in Tswana, Blackeye, and ER7. This validation was performed to genetically eliminate any false resistance that could have arisen from environmental biases [26,27]. The SNP1_0912 marker, which flanks the significant aphid resistance QTL (QAc-vu7.1) in IT97K-556-6 and SARI-2KTA-6 [33,57], confirmed the monogenic homozygous significant aphid resistance QTL (QAc-vu7.1). This marker confirmed aphid resistance in Tswana and ER7. Despite the confirmed resistance in IT97K-556-6, a contrary aphid susceptibility phenotype was observed in this variety. This suggests that, although the presence of the QAc-vu7.1 marker region, as indicated by the SNP1_0912 marker, is confirmed, phenotypic variation may indicate the presence of functional gene loci or QTL loci linked to resistance in Tswana and ER7. Furthermore, this suggested the significant negative effect of the QTL-by-environment interaction effect on the expression of QAc-vu7.1. Meanwhile, the SNP1_0912 marker is located at 21.7 cM, whereas the QTL is at 22 cM on chromosome 2 of the cowpea genome. Despite its closeness to the QTL, this marker might have been ineffective in identifying genetic variation of the phenotypic trait among the cowpea genotypes due to its non-functionality [33,57]. Similar results were observed when using CP 171F/172R, which confers resistance in SARI-2KTA-6. This suggested that, beyond the probable negative effect of the QTL-by-environment interaction and the non-functionality of the SNP marker, the substantially low level of aphid population and high plant vigour in Tswana and ER7 were causative of a probable unknown functional aphid resistance marker region for the specific aphid biotype, and that marker might be substantially expressed under high and dry temperate growth conditions.

## 4. Materials and Methods

### 4.1. Experimental Site

The experiment was conducted in a screenhouse at the Botswana University of Agriculture and Natural Resources Gardens in Sebele during the 2023/2024 growing season. This site is located at a latitude of 24°33′ S and a longitude of 25°54′ E in Sebele, Gaborone, in the southern part of Botswana. The experiment period was between 1 February and March 2024. The minimum temperature recorded during this period was 16.1 °C, the maximum was 40.6 °C, and the average was 20 °C (Meteorological Services Botswana). The average rainfall for the experimental period was 3.8 mm (Meteorological Services Botswana).

### 4.2. Planting Materials and Experimental Design

A total of 12 genotypes were used in the current study. This includes the local landraces B013-F, B261-B, B301, B339, and B359, as well as the farmers’ preferred commercial varieties, Tswana, Blackeye, and ER7, sourced from the Botswana National Plant Genetic Resources Centre. The landraces B261-B, B301, B339, B359, Blackeye, and Tswana were selected based on their performance against aphid damages in the previous studies [28,29,30]. This study also considered the use of West African aphid resistance releases, which are available through the Kirkhouse Trust network of the African Cowpea Breeding program. These include aphid-resistant breeding lines IT97K-556-6 and KVX295-2-124-99 from the International Institute of Tropical Agriculture (IITA), Ibadan, Nigeria. Furthermore, aphid-resistant breeding lines SARI-21KTA-6 (Zaa/566/SARC) and SARC 1-57-2 were sourced from the Council for Scientific and Industrial Research-Savannah Agriculture Research Institute (CSIR-SARI) in Ghana.

The experiment was laid out in a completely randomized design with three replications. The aphid-resistant varieties IT97K-556-6, KVX295-2-124-99, SARI-21KTA-6 (Zaa/566/SARC), and SARC 1-57-2 were used as controls. Two seeds were sown and planted in 2.5 L polyethylene planting bags filled with soil- and later thinned to one plant per potting bag. The chemical composition of the soil was as follows: organic carbon (3.07%), phosphorus (17.21 mg/kg), calcium (19.54 Cmol/kg), magnesium (7.50 Cmol/kg), potassium (2.08 Cmol/kg), and a pH of 6.43.

### 4.3. Aphid Culture

Aphids were collected from a cowpea field near Gaborone and cultured on the susceptible Blackeye in an insect-proof cage. Cowpea seedlings (seven days after emergence) were infested with fourth-instar nymphs (4-to-5-day-old aphids). Five of these apterygote aphids were collected with a camel hairbrush and carefully placed on each plant (Togola et al., 2020 [14,39]). Watering was conducted carefully to prevent aphids from being washed off the plants. The pots remained in the insect-proof cages for 27 days. At the end of the experiment, all the infested plants were sprayed with Dimethoate.

### 4.4. Aphid Infestation and Plant Damage Severity Score

The number of live nymphs, the number of live adults, and the number of alates were recorded at 2-day intervals for 27 days after infestation. Aphids were visually inspected on the leaves, petioles, and stems using a magnifying glass, while counting was performed with the aid of a digital counter (Figure A1). At the end of the experiment, visual aphid plant damage scoring was conducted using a scale of 1 to 5 [36]. Score 1 was for plants with no aphid and no aphid damage, labelled as highly resistant; 2 was for plants with aphids and slight signs of damage (slight yellowing of lower leaves without capping), labelled as resistant; 3 was for plants showing symptoms of aphids’ damage (yellowing of lower leaves and slight capping), labelled as moderately resistant; 4 was for plants with weak stems and leaves with symptoms of aphid damage (severe capping of leaves, stunted plants, yellowing of all leaves), labelled as moderately susceptible; and score 5 indicated dead plants due to aphids’ damage, categorized as highly susceptible [14,36,40].

### 4.5. Marker-Assisted Polymorphic Tests

Local farmers’ preferred commercial cowpea varieties, including Tswana, Blackeye, and ER7, were further subjected to polymorphic tests. In addition to farmers’ preference and the need for improvement of these varieties against biotic stresses, the selection was also based on the phenotypic response to aphids’ damage. The phenotypic resistance and susceptibility of these varieties were validated using the West African aphid-released variety IT97K-556-6. The codominant SNP1_0912 marker (Table 4), which is a flanking marker of the major aphid resistance QTL (QAc-vu7.1) in IT97K-556-6 [33] was used to confirm resistant (165 bp) and susceptible plants (150 bp). In addition, polymorphic tests were conducted on Tswana and ER7 using SARI-2KTA-6 as the control. The SARI-2KTA-6 was developed by pyramiding aphid resistance from IT97K-556-6 and SARC 1-57-2 into a susceptible Zaayura [32]. Therefore, the simple sequence repeat (SSR) marker CP 171F/172R, which flanks aphid resistance in IT97K-556-6, SARC 1-57-2, and SARI-2KTA-6 breeding lines, was used for polymorphic tests.

Total genomic DNA (gDNA) was extracted from six replicates of liquid nitrogen-grounded leaf tissue samples using Quick-DNATM Plant/Seed Miniprep Kit Cat No; D6020 (Zymo Research Corporation, Irvine, CA, USA) as per the manufacturer’s instructions. The quality and concentration of the extracted gDNA were checked on Nanodrop (Nanodrop One, Thermofisher Scientific 5225 Verona Rd. Madison, WI, USA) and a 2.5% agarose gel consisting of 2.5 g agarose, 100 mL of ×1 TAE buffer and 5 μL of ethidium bromide. The polymerase chain (PCR) reaction (PCR) was performed in a Nexus Eppendorf Master Cycler (Eppendorf AG 700W 22331 Hamburg, Germany). The 25 µL PCR reaction consisted of the following: 1X Q5^®^ High-Fidelity reaction buffer, 200 200 µM dNTPs, 0.5 µM of each Forward and reverse primer, 0.02 U/µL Q5 High-Fidelity DNA Polymerase, 1 µg DNA template, and nuclease-free water. Thermal cycling was followed by an initial denaturing at 98 °C for 30 s, denaturing at 98 °C for 10 s, and 35 cycles of annealing at 52 °C for 30 s. The extension was performed at 72 °C for 30 s, the final extension at 72 °C for 2 min, and storage was conducted at 4 °C. The amplicons were separated by agarose gel (2.5% *w*/*v*) electrophoresis in 1X TAE (Tris base-acetic acid-EDTA) buffer at 80 V for 60 min. Gels were stained with ethidium bromide (0.002%, *v*/*v*). A 1 kilobase DNA molecular weight marker (Fast DNA ladder (New England Biolabs Inc., Ipswich, MA, USA) was used. The gels were visualized in a Molecular imager Bio-Rad Gel DocTM XR + System (Biorad Laboratories, Hercules, CA, USA).

### 4.6. Data Statistical Analysis

The data were subjected to one-way analysis of variance (ANOVA) using R software version 4.2.2, using the Agricolae package version 1.3-5. The Fisher’s least significant difference (LSD) test was used to separate the means at a significance level of 5%.

## 5. Conclusions

In this study, we evaluated cowpea genotypes for aphid resistance under screehouse conditions in an insect-proof cage. Surprisingly, IT97K-556-6, SARC1-57-2, and SARI-2KTA-6 were susceptible to aphid damage. Their aphid damage susceptibility could be attributed to several factors, including the negative effect of QTL-by-environment interaction and the possible existence of different aphid biotypes. Tswana exhibited resistance to aphids, probably through the antifeedant property that suppressed the reproductive rate and multiplication of aphids, ultimately leading to their death. However, future studies should consider various mechanisms linked to antixenosis, such as the salivary-related transmission of the virus beyond the epidermis and mesophyll, aphid feeding-induced phytohormones-mediated plant defence, and aphid salivary effector-triggered immunity. The resistance and tolerance to aphids observed in Tswana, B261-B, B301, and B359 provide a basis for marker-assisted breeding in developing genotypes resistant to aphids specific to Botswana’s cowpea aphid biotypes. Tswana might be used in breeding programs involving biparental lines between Tswana as the resistant donor and Blackeye as the susceptible recurrent line. Sequencing the biparental F2 progeny will be crucial in identifying novel functional markers that flank resistance to aphids. Furthermore, these findings provide an important basis for exploring the metabolic and molecular mechanisms to enhance understanding of the function of secondary metabolites and effector-triggered immunity.

## Figures and Tables

**Figure 1 ijms-26-04406-f001:**
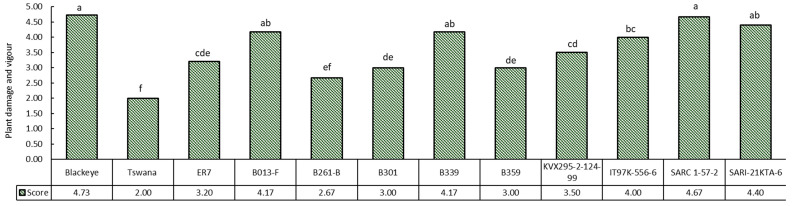
Plant damage and vigour scores at 27 days after infestation. The score rating represented by the different lowercase letters within the graph columns denotes a significant difference between genotypes at *p* < 0.05 (Fishers’ LSD); Score rating of 1: highly resistant; score rating of 2: resistance genotypes; score rating of 3: tolerant phenotype; score rating of 4: moderately susceptible; score rating of 5: highly susceptible.

**Figure 2 ijms-26-04406-f002:**
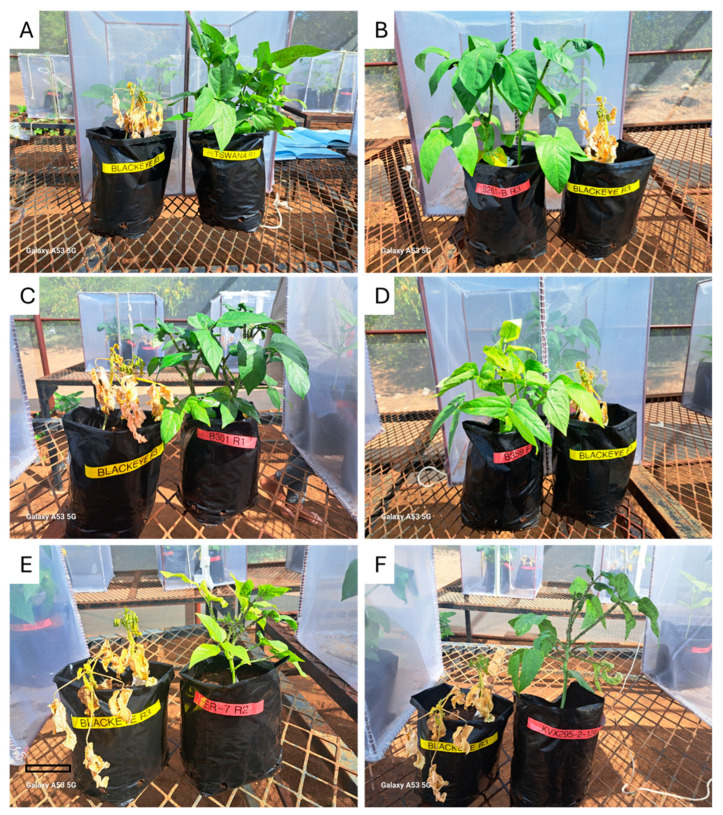
Resistant and tolerant cowpea plant vigour phenotype. The plant vigour for all genotypes was compared to Blackeye. Resistant phenotype: Tswana (**A**) and B261 (**B**); tolerant phenotypes: B301 (**C**), B359 (**D**), ER7 (**E**), and KVX295-2-124-9 (**F**).

**Figure 3 ijms-26-04406-f003:**
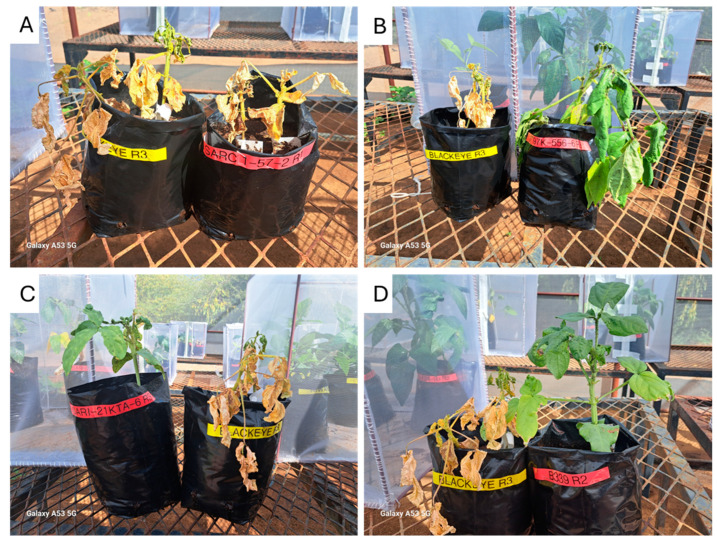
Susceptible cowpea plant vigour phenotype following aphid infestation. Susceptible genotypes include SARC 1-57-2 (**A**), IT97K-556-6 (**B**), SARI-21KTA-6 (**C**), and B339 (**D**).

**Figure 4 ijms-26-04406-f004:**
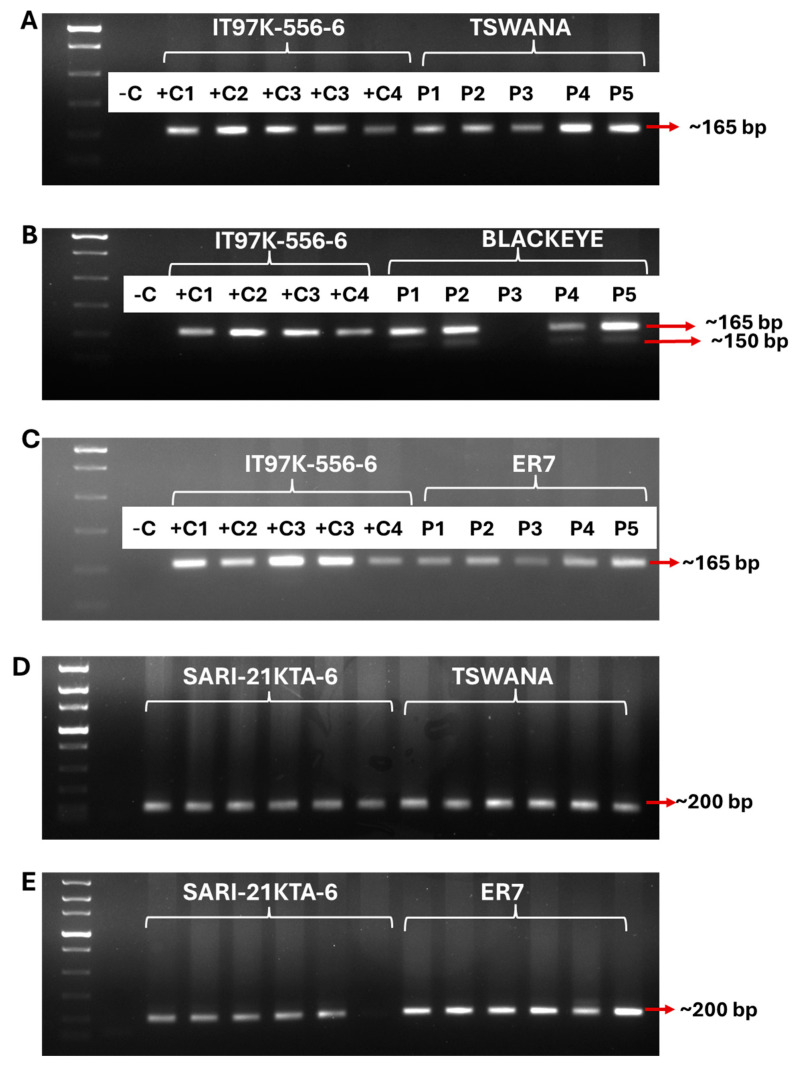
Genotyping of cowpea genotypes for aphid resistance. SNP1_0912 (**A**–**C**) molecular marker, which flanks the major QTL for aphid resistance in IT97K-556-6, was used to validate aphid resistance or susceptibility in Tswana (**A**), Blackeye (**B**), and ER7 (**C**). Resistance in Tswana (**D**) and ER7 (**E**) was further validated using the CP 171F/172R marker, which confers aphid resistance in SARI-21KTA-6. A 2.5% Agarose gel and a 1 kb molecular weight were used. −C: negative control without DNA template; +C1 to +C6: positive control plants conferring aphid resistance; P1 to P6: the targeted plants.

**Table 1 ijms-26-04406-t001:** The number of adult aphids from 7 to 27 days after infestation.

Days After Infestation											
Genotypes	7	9	11	13	15	17	19	21	23	25	27
Tswana	5.3 ± 0.6 ^a^	4 ± 0.0 ^bc^	47.5 ± 11.5 ^abc^	93.3 ± 19.7 ^bc^	57.7 ± 36.3 ^b^	119 ± 100.8 ^a^	118 ± 58.1 ^a^	101.7 ± 51.0 ^a^	94.8 ± 69.8 ^a^	40.8 ± 26.6 ^a^	37.5 ± 16.7 ^a^
Blackeye	4.7 ± 0.6 ^a^	4.5 ± 0.5 ^bc^	31.5 ± 23.3 ^cd^	115.7 ± 58.7 ^abc^	179.8 ± 89.0 ^b^	269.5 ± 168.1 ^a^	288.7 ± 245.4 ^a^	102.8 ± 4.6 ^a^	34.8 ± 24.4 ^a^	23.3 ± 15.9 ^a^	27.5 ± 19.3 ^a^
ER7	5.8 ± 3.8 ^a^	4.3 ± 1.2 ^bc^	13.5 ± 11.2 ^d^	44.8 ± 32.3 ^c^	100.3 ± 57.5 ^b^	140.5 ± 101.2 ^a^	137.2 ± 41.9 ^a^	127.3 ± 78.2 ^a^	54.7 ± 35.5 ^a^	33.7 ± 34.8 ^a^	28.7 ± 29.7 ^a^
B013-F	4.7 ± 0.8 ^a^	5.2 ± 1.3 ^b^	38.2 ± 14.2 ^bcd^	101.7 ± 47.2 ^abc^	177.5 ± 132.4 ^ab^	316.2 ± 148.2 ^a^	129.2 ± 108.0 ^a^	55.5 ± 9.9 ^a^	52.0 ± 31.1 ^a^	20.2 ± 5.5 ^a^	20.3 ± 16.4 ^a^
B261-B	4.8 ± 0.3 ^a^	3.5 ± 0.0 ^bc^	70.2 ± 20.7 ^a^	147.2 ± 36.3 ^ab^	251.7 ± 29.1 ^ab^	364.0 ± 205.8 ^a^	210.2 ± 108.0 ^a^	125.0 ± 34.6 ^a^	32.8 ± 5.5 ^a^	37.7 ± 11.3 ^a^	79.3 ± 26.5 ^a^
B301	5.0 ± 0.0 ^a^	5 ± 0.0 ^bc^	64.8 ± 16.6 ^ab^	93.3 ± 38.8 ^bc^	173.5 ± 77.0 ^b^	206.5 ± 102.4 ^a^	136.3 ± 50.5 ^a^	73.8 ± 23.4 ^a^	28.3 ± 9.1 ^a^	37.8 ± 5.5 ^a^	76.4 ± 1.9 ^a^
B339	4.2 ± 0.6 ^a^	4 ± 0.87 ^bc^	41.7 ± 13.9 ^abcd^	121.8 ± 58.8 ^abc^	156.5 ± 109.6 ^b^	139.5 ± 87.3 ^a^	128.2 ± 54.0 ^a^	39.2 ± 29.2 ^a^	37.7 ± 5.5 ^a^	30.8 ± 5.3 ^a^	41.8 ± 29.6 ^a^
B359	4.8 ± 0.3 ^a^	3.2 ± 1.0 ^c^	31.7 ± 6.0 ^cd^	80 ± 19.1 ^bc^	177.2 ± 87.2 ^ab^	248.2 ± 221.0 ^a^	187 ± 197.7 ^a^	101.3 ± 60.9 ^a^	66.2 ± 51.1 ^a^	49.8 ± 39.3 ^a^	51 ± 42.4 ^a^
SARC 1-57-2	4.8 ± 1.2 ^a^	3.8 ± 1.5 ^bc^	40.7 ± 21.5 ^abcd^	189 ± 95.0 ^a^	198.3 ± 100.6 ^ab^	195.2 ± 53.8 ^a^	124.2 ± 31.6 ^a^	53.8 ± 12.4 ^a^	83.8 ± 12.4 ^a^	38.5 ± 0.0 ^a^	25.5 ± 0.0 ^a^
SARI-2KTA-6	4.5 ± 1.3 ^a^	7 ± 2.6 ^a^	26.3 ± 8.1 ^cd^	109.3 ± 44.6 ^abc^	244 ± 56.6 ^ab^	305 ± 134.3 ^a^	184.2 ± 72.4 ^a^	107 ± 30.0 ^a^	68.5 ± 54.9 ^a^	34.5 ± 21.9 ^a^	46.5 ± 19.1 ^a^
IT97K-556-6	4.3 ± 1.2 ^a^	4.8 ± 0.3 ^bc^	49.3 ± 13.6 ^abc^	192.2 ± 37.4 ^a^	255.3 ± 165.4 ^ab^	366.5 ± 213.1 ^a^	324.8 ± 170.8 ^a^	55.0 ± 37.7 ^a^	112.7 ± 47.6 ^a^	33.5 ± 2.1 ^a^	21.0 ± 21.2 ^a^
KVX295-2-124-99	4.8 ± 0.3 ^a^	4.8 ± 0.3 ^bc^	32.3 ± 17.9 ^cd^	100.3 ± 32.5 ^abc^	377.8 ± 143.2 ^a^	343.8 ± 277.9 ^a^	241.7 ± 188.0 ^a^	145.7 ± 115.8 ^a^	72.5 ± 45.9 ^a^	48.3 ± 45.9 ^a^	44.7 ± 47.1 ^a^
*p*-value	0.95 ns	0.05 *	0.01 **	0.05 *	0.09 ns	0.56 ns	0.60 ns	0.42 ns	0.32 ns	0.96 ns	0.25 ns
F-value	0.38	2.96	2.96	2.3	1.89	0.89	0.85	1.09	1.25	0.34	1.34
CV%	25.7	22.21	39.2	42.1	52.47	65.96	71.1	60.02	66.57	66.57	64.15

*, **, ns, Significant at *p* ≤ 0.05, 0.01, or non-significant, respectively, within columns. Shown values are the means. Means followed by a dissimilar letter within a column are significant at 5% Fisher’s LSD.

**Table 2 ijms-26-04406-t002:** The number of nymphs from 7 to 27 days after infestation.

	Days After Infestation										
Genotypes	7	9	11	13	15	17	19	21	23	25	27
Tswana	39.5 ± 27.8 ^abc^	94.5 ± 38.6 ^abc^	137.8 ± 11.5 ^ab^	240.2 ± 75.7 ^ab^	210.2 ± 151.6 ^c^	248.5 ± 136.0 ^b^	312.3 ± 40.0 ^b^	329.0 ± 320.1 ^b^	267.7 ± 189.7 ^b^	325.2 ± 150.5 ^a^	386.8 ± 67.7 ^abc^
Blackeye	32.2 ± 17.9 ^bcd^	104.5 ± 56.9 ^abc^	89.3 ± 35.2 ^b^	347.7 ± 207.6 ^ab^	767.2 ± 486.3 ^abc^	914.7 ± 564.6 ^ab^	1150.8 ± 883.4 ^ab^	1056.5 ± 241.1 ^a^	1016.5 ± 407.3 ^ab^	538.8 ± 54.8 ^a^	650.3 ± 464.9 ^a^
ER7	8.0 ± 7.0 ^d^	57.8 ± 39.1 ^c^	121.5 ± 60.0 ^ab^	135.0 ± 42.7 ^b^	333.2 ± 198.5 ^bc^	543.8 ± 351.2 ^ab^	649.5 ± 494.0 ^b^	655.7 ± 356.2 ^ab^	309.7 ± 223.0 ^b^	374.0 ± 288.2 ^a^	137.7 ± 139.3 ^bc^
B013-F	31.3 ± 9.8 ^bcd^	102.3 ± 35.1 ^abc^	116.3 ± 22.9 ^ab^	486.8 ± 452.6 ^ab^	609.7 ± 620.6 ^abc^	981.5 ± 807.8 ^ab^	587.2 ± 645.8 ^b^	431.5 ± 75.7 ^ab^	233.3 ± 138.9 ^b^	160.3 ± 24.5 ^a^	94.2 ± 62.6 ^c^
B261-B	58.3 ± 20.0 ^ab^	85 ± 28.5 ^abc^	217.2 ± 127.9 ^a^	696.3 ± 284.7 ^a^	1059 ± 545.1 ^ab^	1672.3 ± 750.3 ^a^	1071 ± 409.0 ^ab^	721.7 ± 168.9 ^ab^	281.2 ± 26.8 ^b^	321.3 ± 121.8 ^a^	184.2 ± 87.4 ^abc^
B301	64.2 ± 16.0 ^a^	149.3 ± 10.5 ^a^	170.8 ± 29.2 ^ab^	333 ± 64.0 ^ab^	727.7 ± 237.9 ^abc^	755.2 ± 348.4 ^ab^	621.3 ± 486.8 ^b^	481.2 ± 277.1 ^ab^	223.2 ± 66.8 ^b^	273.7 ± 59.1 ^a^	285 ± 24.8 ^abc^
B339	23.7 ± 9.1 ^cd^	100.8 ± 26.4 ^abc^	183.8 ± 75.5 ^ab^	350.7 ± 180 ^ab^	487.5 ± 283.3 ^abc^	665 ± 383.2 ^ab^	854.8 ± 481.0 ^ab^	751.5 ± 354.3 ^ab^	771.5 ± 348.6 ^ab^	393.8 ± 347.5 ^a^	173.2 ± 120.9 ^bc^
B359	32.0 ± 9.5 ^bcd^	78.7 ± 28.0 ^bc^	95.3 ± 25.0 ^b^	416.2 ± 226.9 ^ab^	571.5 ± 347.4 ^abc^	1672.3 ± 750.3 ^a^	498.5 ± 447.5 ^b^	435.3 ± 301.3 ^ab^	250.8 ± 128.0 ^b^	390.7 ± 261.6 ^a^	501 ± 343.7 ^abc^
SARC 1-57-2	28.7 ± 18.6 ^bcd^	104.8 ± 36.8 ^abc^	159.2 ± 21.3 ^ab^	407.0 ± 229.9 ^ab^	484.3 ± 211.9 ^abc^	1055.2 ± 231.0 ^ab^	867.0 ± 254.6 ^ab^	361.3 ± 111.4 ^b^	375.8 ± 25.1 ^b^	481.0 ± 0.0 ^a^	322.0 ± 0.0 ^abc^
SARI-2KTA-6	28.8 ± 7.3 ^bcd^	132.8 ± 17.5 ^ab^	104.3 ± 17.2 ^b^	482.7 ± 235.3 ^ab^	924.0 ± 148.6 ^abc^	1184.3 ± 87.7 ^ab^	1097.2 ± 331.4 ^ab^	576.3 ± 85.8 ^ab^	648.5 ± 384.7 ^ab^	411.7 ± 209.4 ^a^	608.5 ± 130.8 ^ab^
IT97K-556-6	38.0 ± 7.7 ^abcd^	139.0 ± 18.5 ^ab^	84.8 ± 67.9 ^b^	551.0 ± 224.1 ^ab^	1163.7 ± 622.7 ^a^	1636.3 ± 881.5 ^a^	1810.3 ± 778.3 ^a^	720.0 ± 550.3 ^ab^	1573.5 ± 977.3 ^a^	629.5 ± 75.7 ^a^	330.5 ± 410.8 ^abc^
KVX295-2-124-99	29.5 ± 20.8 ^bcd^	82.2 ± 32.8 ^bc^	116.0 ± 42.3 ^ab^	539.8 ± 431.4 ^ab^	886.3 ± 323.4 ^abc^	947.5 ± 700.8 ^ab^	1021.3 ± 743.4 ^ab^	683.5 ± 421.5 ^ab^	1127.2 ± 814.0 ^ab^	688.5 ± 878.7 ^a^	557.8 ± 504.5 ^abc^
*p*-value	0.03 *	0.1 ns	0.09 ns	0.50 ns	0.20 ns	0.16 ns	0.22 ns	0.38 ns	0.03 *	0.83 ns	0.09 ns
F-value	2.59	1.86	1.88	0.97	1.55	1.8	1.45	1.15	2.61	0.57	2.04
CV%	46.26	32.9	39.4	63.42	58.85	57.57	64.73	52.43	78	79.91	62.1

*, ns, Significant at *p* ≤ 0.05, 0.01, 0.001, or non-significant, respectively, within columns. Shown values are the means. Means followed by a dissimilar letter within a column are significant at 5% Fisher’s LSD.

**Table 3 ijms-26-04406-t003:** The average number of Alates between 19 and 27 days after aphid infestation.

	Days After Infestation				
Genotypes	19	21	23	25	27
Tswana	57.8 ± 54.5 ^a^	41.3 ± 12.2 ^b^	48.7 ± 23.2 ^b^	45.5 ± 30.5 ^a^	54.0 ± 30.2 ^bc^
Blackeye	117.0 ± 31.8 ^a^	260.5 ± 171.8 ^a^	219.3 ± 20.9 ^a^	116.3 ± 39.2 ^a^	174 ± 116.0 ^abc^
ER7	22.2 ± 12.2 ^a^	67.3 ± 38.6 ^b^	131 ± 126.3 ^ab^	68.7 ± 70.9 ^a^	57.7 ± 64.3 ^bc^
B013-F	44.5 ± 36.3 ^a^	66.5 ± 12.7 ^b^	131.3 ± 132.6 ^ab^	44.5 ± 23.0 ^a^	26.0 ± 20.8 ^c^
B261-B	94.2 ± 53.5 ^a^	62 ± 13.5 ^b^	53.5 ± 4.8 ^b^	92.2 ± 16.8 ^a^	75.8 ± 10.3 ^bc^
B301	125.8 ± 67.2 ^a^	90.2 ± 35.3 ^b^	148 ± 92.2 ^ab^	101.5 ± 79.0 ^a^	96 ± 22.6 ^bc^
B339	97.5 ± 67.5 ^a^	101 ± 29.1 ^b^	99.3 ± 24.4 ^ab^	55.75 ± 9.5 ^a^	53 ± 20.1 ^bc^
B359	66.7 ± 55.4 ^a^	91.5 ± 44.0 ^b^	83.7 ± 78.9 ^ab^	83 ± 68.2 ^a^	33 ± 31.4 ^c^
SARC 1-57-2	111.5 ± 103.0 ^a^	157.5 ± 60.1 ^ab^	173.8 ± 87.3 ^ab^	134 ± 0.0 ^a^	63 ± 0.0 ^bc^
SARI-2KTA-6	92 ± 45.0 ^a^	143.7 ± 72.4 ^ab^	190 ± 29.5 ^ab^	107.33 ± 39.9 ^a^	272.8 ± 6.0 ^a^
IT97K-556-6	113.2 ± 40.6 ^a^	171.7 ± 108.7 ^ab^	169.7 ± 48.0 ^ab^	112 ± 10.0 ^a^	91.5 ± 113.8 ^bc^
KVX295-2-124-99	76 ± 86.7 ^a^	117.5 ± 80.0 ^b^	131 ± 96.8 ^ab^	81.2 ± 68.0 ^a^	189.2 ± 177.3 ^ab^
*p*-value	0.57 ns	0.08 ns	0.27 ns	0.48 ns	0.03 *
F-value	0.89	2.06	1.35	0.67	2.66
*CV%*	69.17	59.79	59.37	62.15	73.92

*, ns: significant at *p* < 0.05, 0.01 or non-significant (ns), respectively, within columns. Shown values are the means. Means followed by a dissimilar letter within a column are significant at 5% Fisher’s LSD.

**Table 4 ijms-26-04406-t004:** Primer sequences for co-dominant SNP1_0912 marker.

Primer Name	Primer Sequence (5′-3′)	Reference
CP 171F/172R	F: GTAGGGAGTTGCCCACGAATA R: CAACCGATGTAAAAAGTGGAC	[58]
SNP1_0912	1_0912R: ACTGTTGGATCCGATTGAGG 1_0912F1: ACCATACATTACATATAACTAACTTCTCGCCATGACT 1_0912F2: ATTAATAACTAACTTCTCGCCATGAGC	[57]

## Data Availability

The original contributions presented in the study are included in this article, and further inquiries can be directed to the corresponding authors.

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
