# Peer review of "Phenotypic Screening and Marker-Assisted Validation of Sources of Aphis craccivora Koch Resistance in Cowpea (Vigna unguiculata L.)"

_ijms, 2025, doi:10.3390/ijms26094406_

Round 1

Reviewer 1 Report

Comments and Suggestions for Authors

Major concerns

  1. Typical photos for aphid damage and plant vigour score should be listed for each variety.
  2. How were the aphids counted? Which should be described in materials and methods part.
  3. Why were the four aphid-resistant varieties susceptible to aphids? They are suitable as susceptible controls. Why varieties with score 1 were not selected as resistant controls?
  4. The SNP marker results revealed that Blackeye was different from IT9K-556-6, Tswana and ER7, which means IT9K-556-6, Tswana and ER7 should be aphid resistant if the marker confers the resistance. Why only Tswana was resistant, without IT9K-556-6 and ER7? Is the marker reliable for conferring the resistance? I don’t understand how the marker confirm resistance.
  5. Why only four varieties were selected for maker analysis? How about the results for the other varieties?
  6. The authors should be more cautious about the antibiosis and antixenosis related to resistance since there are no direct proofs. They are just hypothesis, which could not be directly indicated by the results. Besides, there are also no proofs for ‘overexpression of the susceptible allele’ in abstract part.

Minor concerns

  1. It should be 12 genotypes in line 212.

Author Response

Thank you very much for taking the time to review this manuscript. We are so grateful for your very important comments that contributed to the enhancement of both knowledge and quality of the manuscript. Please find detailed responses that have been revised/corrected in the submitted manuscript. Major concerns 1. Typical photos for aphid damage and plant vigour score should be listed for each variety. • Agree, and this has been included in Figure 2 and 3 respectively. • Information on the respective figures has been included under plant damage and vigour results 2. How were the aphids counted? Which should be described in materials and methods part. • Aphids were visualised using magnifying glass while counting was done on leaves, petiole, and stems of each plant using digital counter 3. Why were the four aphid-resistant varieties susceptible to aphids? They are suitable as susceptible controls. Why varieties with score 1 were not selected as resistant controls? • Susceptibility of the aphid resistance released varieties might be linked to various factors, such as the environmental-QTL interaction, possibly leading to the non-functionality of the marker. Secondly, this may be due to the prevalence of a different aphid biotype in Southern Africa (Botswana), for which resistance is not apparent in West African varieties. • We agree that after conducting phenotypic tests under conditions in Botswana, these genotypes can be recommended as suitable susceptible controls. However, based on the literature regarding the marker region associated with resistance to aphids, the study used these genotypes as resistant controls and established a genetic difference between them and the local genotypes. Our results indicated an inverse relationship between the identified markers and plant vigour. Therefore, these markers may not be linked to aphid resistance in the local aphid biotype in Botswana, although they do function for aphid biotypes found in Western Africa.. • There was no variety with a score 1 (a highly resistant variety) as per the method elaborated under methods on ratings. Scoring was based on 1 to 5 atings; 1-highly resistant, 2-resistant, 3 tolerant, 4-moderately susceptible while 5 is for highly susceptible. • In terms of selections, IT97K-556-6 and SARI (added gels) with a score rating of 4 (susceptible) were included for the polymorphic tests and gels are included in Figure 4. 4. The SNP marker results revealed that Blackeye was different from IT9K-556-6, Tswana and ER7, which means IT9K-556-6, Tswana and ER7 should be aphid resistant if the marker confers the resistance. Why only Tswana was resistant, without IT9K-556-6 and ER7? Is the marker reliable for conferring the resistance? I don’t understand how the marker confirm resistance. • Agree, that SNP marker showed Blackeye was different from Tswana, ER7 and IT97K-556-6. The marker confirmed the presence of resistance that is present in IT97K-556-6. However, the contrary phenotypic results showed IT97K-556-6 to be susceptible. This contradicting observation suggested that the marker responsible for aphid resistance in IT95K-556-6 was not linked to aphid resistance in Tswana and ER7. Similarly, the CP markers had consistent results. Altogether, this suggests that they are novel functional gene loci or quantitative trait loci (QTL) associated with aphid resistance in Tswana and ER7. • This is a niche for further studies to unravel the genetic mechanisms linked to aphid resistance in Tswana and ER7 and further identify and develop markers. Those markers may be reliable under Southern African conditions and the specific aphid biotype. 5. Why only four varieties were selected for maker analysis? How about the results for the other varieties? • This were selected based on farmer’s preference and the need for improvement for biotic stresses as part of the smart agriculture strategy in Botswana. 6. The authors should be more cautious about the antibiosis and antixenosis related to resistance since there are no direct proofs. They are just hypothesis, which could not be directly indicated by the results. Besides, there are also no proofs for ‘overexpression of the susceptible allele’ in abstract part. • Fully agree that no data linked to other factors associated with these mechanisms. However, the write up considered the use of words that indicate the potential possibilities linked to those. Minor concerns 1. It should be 12 genotypes in line 212. • Corrected

Reviewer 2 Report

Comments and Suggestions for Authors

Overall, the work is exciting. I recommend, however, describing in more detail the genotypes used their provenance and why they were chosen in such a way as to explain in the results obtained the differences found.in suggest to review the language and also to improve the references.

Author Response

Thank you very much for taking the time to review this manuscript. We are so grateful for your very important comments that contributed to the enhancement of both knowledge and quality of the manuscript. Please find detailed responses that have been revised/corrected in the submitted manuscript. Overall, the work is exciting. I recommend, however, describing in more detail the genotypes used their provenance and why they were chosen in such a way as to explain in the results obtained the differences found.in suggest to review the language and also to improve the references. • Descriptions of the genotypes and their origins has been revised under materials. • Selection of the landraces was based on previous screening methods, which indicated that they showed an aphid resistance phenotype. This was also revised in the introduction by adding information on the previous studies, which justifies the need to identify or confirm their inherent genetic mode of aphid resistance. • This project is an initiative of the Kirkhouse Foundation under the Cowpea Improvement Project. Therefore network is currently using these breeding lines to improve cowpea elite lines in African region. These are commonly from West Africa since the program was initially targeting West Africa. Furthermore, there are no information on aphid resistance markers for Southern Africa. • The language for the write up has been improved by the use of the premium services of Grammarly • In terms of References, it is agreed that they were discrepancies especially on scientific names and names of journals. This has been revised accordingly and uniformity of the write up has been followed.

Reviewer 3 Report

Comments and Suggestions for Authors

Author Response

Thank you very much for taking the time to review this manuscript. We are so grateful for your very important comments that contributed to the enhancement of both knowledge and quality of the manuscript. Please find detailed responses that have been revised/corrected in the submitted manuscript. Comments and Suggestions for Authors Line 16: Aphis craccivora in italics: Corrected/italicized Line 45: delete: Deleted Line 61: Vigna unguiculata → V. unguiculata in italics: Corrected/italicized Line 66: delete Cowpea: Deleted Line 80: please check IOCV rules for virus naming: naming corrected Line 195: un-italicize Figure 2.: Corrected/un-italicized Line 196: double-check Line 196 for any errors/typos: “allele is M” has been removed from the sentence. Line 228: use un-bold font: unbolded Line 239: Description for Score 1 needs to be revised. • The score ratings has been described score ratings adopted from authors (Togola et al., 2020; Kityo et al., 2021; Seram and & Devi 2021). A detailed description of the method has been added. • However, it is agreed that in our opinion that more robust method and description in that respect need to be revised Line 248: three local genotypes? Line 213 stated these three are commercial varieties from Botswana National Plant Genetic Resource Center. Clarification needed. • They (Tswana, blackeye, and ER7) are the most commonly produced varieties in Botswana and they possess desirable traits that make them suitable for widespread production and marketability in the country. There are also under the country list of priority for smart agriculture and the need for improvement for both biotic and abiotic stresses. Line 250: SARC 1-57-?? • Typo Corrected (SARC 1-57-2) Line 251: the local varieties? see the comment for Line 248. • Addressed Line 261: 2% gel? Line 195 said 2.5%. • 2.5% gel was used for both gDNA integrity and PCR amplicons Line 262-263: RT reaction with gDNA? Revision needed. • Agree, It was supposed to be PCR not RT-PCR. Typo corrected Line 264: Add the recipe for PCR mixture • Agree, The PCR mixture has been revised and its detailed Line 266: PCR parameters need to be corrected. • Agree, the whole sentence was amended and written in more detail as per the polymerase instructions and the primers’ annealing temperature Line 267: 1.5% gel? Which one was used in the study? See the comment for Line 261 • 2.5% gel was used and all has been corrected Line 269: 10kilobase ladder? • Agree, it was a 1 kb ladder Line 270: was the ladder “molecular weight” marker? • The ladder refers to molecular weight marker and this has been amended to the the use of the latter to avoid confusion Line 281: Reproduction … is through pathogenesis(?) • This was a typo and has been corrected to parthenogenesis Line 281-283: Revision recommended • . Agree, the whole sentence has been revised and long sentences avoided to give a meaningful information. Line 283-286: It is a long and confusing sentence. Revision needed: .• Agree, and revised as mentioned above Line 305: Resistance: corrected

Round 2

Reviewer 1 Report

Comments and Suggestions for Authors

The authors have addressed most of the comment. The manuscript could be accepted after grammar checking.